# A Disposable Electrochemical Biosensor Based on Screen-Printed Carbon Electrodes Modified with Silver Nanowires/HPMC/Chitosan/Urease for the Detection of Mercury (II) in Water

**DOI:** 10.3390/bios11100351

**Published:** 2021-09-23

**Authors:** Apichart Saenchoopa, Supannika Klangphukhiew, Rachata Somsub, Chanon Talodthaisong, Rina Patramanon, Jureerut Daduang, Sakda Daduang, Sirinan Kulchat

**Affiliations:** 1Department of Chemistry, Faculty of Science, Khon Kaen University, Khon Kaen 40002, Thailand; apichats@kkumail.com (A.S.); chanon@kkumail.com (C.T.); 2Department of Biochemistry, Faculty of Science, Khon Kaen University, Khon Kaen 40002, Thailand; supannika.k@kkumail.com (S.K.); narin@kku.ac.th (R.P.); 3Department of Physics, Faculty of Science, Khon Kaen University, Khon Kaen 40002, Thailand; petch.rachataa@gmail.com; 4Department of Clinical Chemistry, Faculty of Associated Medical Sciences, Khon Kaen University, Khon Kaen 40002, Thailand; jurpoo@kku.ac.th; 5Division of Pharmacognosy and Toxicology, Faculty of Pharmaceutical Sciences, Khon Kaen University, Khon Kaen 40002, Thailand

**Keywords:** silver nanowires, screen-printed carbon electrode, urease, electrochemical sensor, portable biosensor

## Abstract

This work describes the facile preparation of a disposable electrochemical biosensor for the detection of Hg(II) in water by modifying the surface of a screen-printed carbon electrode (SPCE). The surface modification consists of the immobilization of a composite layer of silver nanowires, hydroxymethyl propyl cellulose, chitosan, and urease (AgNWs/HPMC/CS/Urease). The presence of the composite was confirmed by scanning electron microscopy (SEM) and its excellent conductivity, due chiefly to the electrical properties of silver nanowires, enhanced the sensitivity of the biosensor. Under optimum conditions, the modified SPCE biosensor showed excellent performance for the detection of Hg(II) ions, with an incubation time of 10 min and a linear sensitivity range of 5–25 µM. The limit of detection (LOD) and limit of quantitation (LOQ) were observed to be 3.94 µM and 6.50 µM, respectively. In addition, the disposable and portable biosensor exhibited excellent recoveries for the detection of Hg(II) ions in commercial drinking water samples (101.62–105.26%). The results are correlated with those obtained from inductively coupled plasma optical emission spectrometry (ICP-OES), indicating that our developed sensor is a reliable method for detection of Hg(II) in real water samples. The developed sensor device is a simple, effective, portable, low cost, and user-friendly platform for real-time detection of heavy metal ions in field measurements with potential for other biomedical applications in the future.

## 1. Introduction

Heavy metal ions cause well-known and extremely severe problems in the environment and in animal and human health [1,2]. Mercury(II) ions are one of the most toxic pollutants and can spread widely in the environment from industrial wastes and natural sources [3,4,5]. Moreover, Hg(II) ions can damage human organs, resulting in serious diseases such as kidney failure and brain and heart damage [6]. The toxicity of Hg(II) ions varies with dose and delivery vector. High exposure to mercury vapor causes pneumonitis, which in extreme cases can be fatal [7]. Mercurous and mercuric salts on the other hand damage the gut lining and kidneys, while methyl mercury can be widely spread throughout the body [7].

Recently, there has been much interest in the development of improved technologies and devices for the detection and monitoring of Hg(II) ions in water, including nanomaterials and nanomaterials-based electrochemical methods [8,9,10,11]. Fast, sensitive, selective, and accurate monitoring of the presence of Hg(II) ions in the environment and drinking water samples is highly desirable [12]. Some examples of well-established detection methods used for Hg(II) detection in water and food in the last decade include atomic fluorescence spectroscopy (AFS) [13], high-performance liquid chromatography (HPLC) [14], flame atomic absorption spectroscopy (FAAS) [15], and graphite furnace atomic absorption spectroscopy (GFAAS) [16]. Although these methods provide the requisite high accuracy and sensitivity, they are costly and require specialist operation. To overcome these problems, functionalized nanomaterials-based sensors incorporating silver nanoparticles (AgNPs [17]), gold nanoparticles (AuNPs [18]), and silver nanowires (AgNWs [19]) have been developed to provide simple, low-cost, and fast sensing of Hg(II) in real samples. In the same direction, electrochemical biosensors have become very efficient point-of-care (POC) devices that can provide fast and highly accurate analysis, and they are portable, easy to use, and low cost [20]. The combination of nanomaterials with electrochemical biosensors for rapid and simple determination of heavy metal ions is an exciting prospect [21].

Electrochemical enzyme inhibition-based biosensors for the detection of heavy metal ions in food and water samples have been widely investigated. For example, electrochemical biosensors for determination of the concentration of Hg(II) based on glucose oxidase and horseradish peroxidase (HRP), L-lactate dehydrogenase, and urease enzymatic inhibition have been constructed [22]. In that work, the quantification of Hg(II) ions was investigated electrochemically by monitoring ammonium ion concentrations produced during an enzymatic reaction using an indirect sensing method [21] and employing the cyclic voltammetry (CV) method. The incorporation of nanomaterials into such biosensors promises improved performance due to their high surface area-to-volume ratio, good electrical properties, and high conductivity [23]. Modification of the electrode surface with nanomaterials improves electrochemical performance, supporting faster electron transfer between a redox probe or electro-active analytes [24,25]. Highly conductive, one-dimensional (1-D) metallic nanowires have received particular attention in this regard [26,27,28]. In this work we employ silver nanowires (AgNWs) composites to modify the surface of a screen-printed carbon electrode (SPCE) towards improved performance and biocompatibility for biosensing of Hg(II).

Specifically, we developed a biosensing platform based on a silver nanowire/hydroxypropyl methylcellulose/chitosan/urease (AgNWs/HPMC/CS/Urease) composite modification to a screen-printed carbon electrode (SPCE) and tested it as a Hg(II) biosensor (Figure 1). The AgNWs/HPMC/CS/Urease layer displays excellent surface conductivity due to the AgNWs, is biodegradable and hydrophilic due the HPMC matrix [29], and has high loading of urease enzyme due to the chitosan adhesive agent [30], increasing the sensitivity and the stability of the modified electrode. We then tested our biosensor, including on commercial drinking water samples, to verify its sensitivity and practicality for the detection of Hg(II) and other metal ions in real world applications. Under optimum conditions, the modified SPCE biosensor exhibited excellent performance with a linear sensitivity range of 5–25 µM for the detection of Hg(II) ions and limit of detection (LOD) and limit of quantitation (LOQ) of 3.94 µM and 6.50 µM, respectively. The biosensor also exhibited excellent recoveries for the detection of Hg(II) ions in commercial drinking water samples (101.62–105.26%).

## 2. Materials and Methods

### 2.1. Materials

Silver nitrate (AgNO_3_, 99.9%) was purchased from POCH™, Poland. Sodium chloride (NaCl, 80%), ethylene glycol (EG) (OHCH_2_CH_2_OH, ≥99.5%), and urea (NH_2_CONH_2_) were purchased from Ajax Finchem, Australia. Polyvinylpyrrolidone (PVP, 360-1000G) and potassium ferricyanide (K_3_Fe(CN_6_), 99%) were purchased from Sigma Aldrich, China. Lead(II) nitrate (Pb(NO_3_)_2_, 99%), nickel(II) nitrate hexahydrate (Ni(NO_3_)_2_·6H_2_O, 99%), potassium chloride (KCl), and cadmium nitrate tetrahydrate (Cd(NO_3_)_2_·4H_2_O, 99%) were purchased from Carlo Erba, Italy. Acetone (CH_3_COOH_3_, ≥99.8%) and ethanol (C_2_H_5_OH, ≥99.7) were purchased from RCI Labscan, Thailand. Copper(II) nitrate trihydrate (Cu(NO_3_)_2_·3H_2_O, ≥99%) and zinc nitrate hexahydrate (Zn(NO_3_)_2_·6H_2_O, ≥99%) were purchased from Fluka, Switzerland. Magnesium nitrate (Mg(NO_3_)_2_, ≥99.4%), sodium nitrate (NaNO_3_, ≥99%), and cobalt(II) nitrate hexahydrate (Co(NO_3_)_2_·6H_2_O, ≥98%) were purchased from Univar, Australia. Calcium chloride (CaCl_2_, ≥95%) was purchased from Scharlau, Spain. Mercury(II) chloride (HgCl_2_, ≥99.5%) was purchased from QRec™, New Zealand. Hydroxypropyl methylcellulose (HPMC) was purchased from Alfa Aesar, China. Chitosan ((C_6_H_11_O_4_)_n_, 100,000–300,000) was purchased from ACRŌS, China. Enzyme urease was prepared from soybeans at the Biochemistry Department, Faculty of Science, Khon Kaen University, Thailand. Deionized water (DI) with a specific resistivity of 18.2 MΩ·cm was obtained from a RiO_s_^TM^ Type I Simplicity 185 (Millipore water purification system).

### 2.2. Instrumentation and Cells

The morphology of silver nanowires was determined using focused ion beam scanning electron microscopes (FIB-SEM, FEI Helios NanoLab G3 CX, Czech Republic). Ultraviolet-visible (UV-Vis) spectra were recorded using an Agilent Technologies Cary 60 UV-visible spectrophotometer (Germany) for absorbance measurement with a 1.0 cm path length quartz cell. Spectra were recorded from 200–800 nm. The detection results were validated with an inductively coupled plasma optical emission spectrometer (PerkinElmer (Wellesley, MA, USA) model OPTIMA 2100). Cyclic voltammetry (CV) was performed using an electrochemical workstation (ECAS100, Zensor Co., Ltd., Taichung, Taiwan). The biosensors were fabricated on screen-printed carbon electrodes (SPCEs, Zensor, Co., Ltd., Taichung, Taiwan) with a three-electrode system: a carbon electrode (d = 3.0 mm/active surface area = 0.071 cm^2^) was used as working electrode, another carbon electrode was used as a counter electrode, and an Ag/AgCl electrode was used as a reference electrode. The NFC potentiostat (NFC microchip SIC4340/41) was obtained from Silicon Craft Technology PLC (Bangkok, Thailand). The NFC potentiostat was operated by the SIC4340/41-POTEN android mobile application equipped on a Motorola One smartphone (Motorola, Chicago, IL, USA).

### 2.3. Synthesis of Silver Nanowires (AgNWs)

AgNWs were synthesized using a modified polyol process [31]. A solution of PVP was prepared by dissolving 0.056 g PVP with 1 mL of ethylene glycol into a 100 mL three-neck, round bottom flask placed in an oil bath and stirred at room temperature. Then, 7.7 mL of ethylene glycol, 0.2 mL of 308.4 mM NaCl, and 0.1 mL of 241.0 mM NaBr were added to the previous solution. The solution was stirred for 10 min and then heated to 170 °C with stirring for 30 min. During heating, nitrogen gas was bubbled through the reaction solution. Then, 1 mL of freshly prepared 265.6 mM AgNO_3_ was added dropwise to the stirring solution and a grey colloid formed. Next, the flask was capped, and the reaction was allowed to take place for 1 h without stirring and heating. After 1 h, the solution was poured into a 1000 mL beaker. Next, the AgNWs were purified by slowly adding 30 mL DI water and 120 mL acetone to the solution. The color of the solution turned to grey-yellow, and the precipitation occurred after 10 min. Next, the supernatant was removed with a pipette. The aggregated AgNWs were redispersed in 20 mL of DI water containing 0.5% *w*/*v* PVP. The redispersed AgNWs were purified again by adding 120 mL of acetone, allowing the AgNWs to settle for 10 min. The solution was centrifuged at a rate of 4000 rpm/min for 10 min and then the supernatant was removed again. Next, ethanol was added to AgNWs for further purification to remove the remaining PVP. The solution was centrifuged again at a rate of 4000 rpm/min for 10 min and the supernatant was removed. Finally, AgNWs were redispersed in 20 mL of ethanol to obtain a concentration of 0.5 g/L and the colloid solution was stored at 4 °C.

### 2.4. The Extraction of Urease

One hundred grams of soybeans were soaked in 400 mL of distilled water in a 500 mL beaker for 24 h. Then, the soaked soybeans were washed with distilled water three times and then 400 mL of distilled water was added. Next, clean soaked soybeans were blended with a blender and filtered to obtain the soybean solution. The soybean solution was centrifuged at a rate of 10,000 rpm/min for 30 min at 25 °C. Finally, the supernatant (crude urease) was collected and then lyophilized using FreeZone 6 Liter Benchtop Freeze Dry Systems with Stoppering Tray Dryers, Labconco. The activity of urease was tested using an acid/base indicator (phenol red) to monitor the enzymatic activities for colorimetric assay [32,33]. The 2.5 µg/µL urease and 2% *w*/*v* urea solution were added to 0.05 mM phenol red as shown in Appendix A. The color of the phenol red solution was changed from yellow to pink, indicating that ammonium species were obtained as the final product [33].

### 2.5. Preparation of Chitosan (CS) and Urease Solution

Firstly, the 2% *w*/*v* chitosan solution was prepared by dissolving 0.4 g of chitosan in 20 mL of 1% *w*/*v* acetic acid while stirring for 30 min. Next, 5 µg/µL urease solution was prepared by dissolving 5 mg urease in 1 mL DI water. Then, the solution of chitosan and urease were mixed using the ratio of 2:5 (chitosan/urease) solution, described as CS/Urease, before coating on the screen-printed carbon electrode (SPCE).

### 2.6. The Fabrication of the AgNWs/HPMC/CS/Urease Modified SPCE

Prior to modification, AgNWs were mixed with 0.05% *w*/*v* hydroxypropyl methyl cellulose (HPMC) in a ratio of 1:1. Then, a 5 μL nanocomposite of AgNWs/HPMC dispersion was deposited on the surface of the working electrode of the SPCE and dried at room temperature for 2 h. After drying the SPCE, 5 μL of chitosan/urease (CS/Urease) solution was immobilized on the deposited nanocomposite AgNWs/HPMC working electrodes and dried at 4 °C. The immobilized enzymatic electrochemical biosensors weredescribed as AgNWs/HPMC/CS/Urease/SPCE. In addition, CS/Urease/SPCE and Urease/SPCE were prepared using a similar method for comparison. A diagram of the device for fabrication is shown in Figure 1. For testing the applicability of our modified electrode, Hg(II) ions were spiked in commercial water samples for investigation in this work. For each concentration and experiment for parameters screening, three electrodes were prepared and all experiments were repeated three times.

## 3. Results and Discussion

### 3.1. Synthesis and Characterization of AgNWs

The silver nanowires (AgNWs) were synthesized via a polyol process [31] and purified following the procedure in Section 2.3. The as-prepared AgNWs were characterized by using UV-vis absorption spectroscopy (Figure 1a). The extinction spectrum shows a dip at 328 nm due to the bulk plasmon of silver, but then strong scattering/absorption at longer wavelengths with a shoulder at 351 nm and a broad peak at 402 nm. The latter correspond to transverse and longitudinal surface plasmon resonances of the silver nanowires, respectively [34]. In addition, the morphology of the AgNWs was examined by scanning electron microscopy (SEM) and transmission electron microscopy (TEM) as depicted in Figure 1b,c. The SEM and TEM images indicated that the AgNWs had a mean diameter of 41.36 ± 8.09 nm and a lattice spacing of 0.204 nm and 0.240 nm. The latter corresponding to the (200) and (111) planes of the silver nanowire, respectively [35].

### 3.2. Fabrication of the Biosensor and Electrochemical Characterization

In this work, a nanocomposite material of AgNWs/HPMC/CS/Urease was coated onto SPCEs by a simple casting method and cyclic voltammograms (CVs) were measured (Figure 2). The electrochemical performance of the modified electrode was compared to a bare SPCE, and to Urease/SPCE, Chitosan/Urease/SPCE, and AgNWs-modified SPCEs. The CVs (Figure 2 and Appendix A) were scanned over the potential range −600 mV to 600 mV with a scan rate of 100 mVs^−1^ using a solution of 5 mM K_3_[Fe(CN)_6_] in 0.1 M KCl as a redox probe. Each electrode type showed well-defined but different redox response. The highest anodic and cathodic peak currents were obtained for the bare SPCE, with a peak potential separation (ΔE_p_) of 145 mV (Appendix A). After the urease enzyme was immobilized on the SPCE, the CV peak current reduced significantly while ΔE_p_ increased to 250 mV, implying the successful immobilization of urease on the SPCE surface (Figure 2, blue line). The peak current response was reduced because the poorly conductive urease layer slowed the electron transfer kinetics at the working electrode. After a mixture of chitosan and urease (CS/Urease) was deposited on the SPCE electrode, the CV peak again showed lower peak current with ΔE_p_ of 231 mV (Figure 2, red line), similar to that seen with Urease/SPCE. For the AgNWs-HPMC/CS/Urease-modified SPCE, however, higher peak currents were observed, with ΔE_p_ of 190 mV (Figure 2, black line). Clearly, the presence of silver nanowires within the immobilized surface coating increased electrochemical response to the K_3_[Fe(CN)_6_] redox probe, and this was attributed both to an increased electroactive surface area and to improved electron transfer kinetics due to the excellent electrical properties of silver nanowires [36,37].

### 3.3. Study of Surface Morphology

The surface morphologies of modified electrodes, including the bare SPCE, AgNWs-modified SPCE, and AgNWs/HPMC/CS/Urease-modified SPCE, were investigated by SEM (Figure 3a–c). A highly interpenetrated network of AgNWs was observed on the AgNWs-modified SPCE surface (Figure 3b and Appendix A). In contrast, the AgNWs/HPMC/CS/Urease-modified SPCE had a smoother surface, with the HPMC, chitosan, and urease solution apparently filling the gaps in the AgNWs network. This may be the ideal scenario, with the network of silver nanowires increasing the electroactive surface area and facilitating electron transfer with proximal adsorbates [38].

### 3.4. Effect of Scan Rate

The electrochemical behavior of the AgNWs/HPMC/CS/Urease-modified electrode was evaluated using a standard solution of K_3_[Fe(CN)_6_] as a redox couple probe. The CV curves of 5 mM K_3_[Fe(CN)_6_]/0.1 KCl were measured at various scan rates ranging from 100 to 800 mV/s by sweeping the potential from −600 to 600 mV in the presence of 20 µM Hg(II) and 2% *w*/*v* urea. The results shown in Figure 4a reveal that the redox peak currents are significantly increased, and the peak positions are slightly shifted by increasing the scan rate, implying that the diffusion layer was decreasing [39]. In addition, a linear relationship between the peak current and the square root of the scan rate was observed, with a correlation coefficient of 0.9806 and 0.9996 for the anodic and cathodic peak currents, respectively (Figure 4b). This indicates that the redox process is diffusion-controlled [40]. Regarding the Randles–Sevcik equation [41], the diffusion coefficient of K_3_[Fe(CN)_6_] was calculated to be 2.4 × 10^−8^ cm^2^/s. A scan rate of 100 mV/s was chosen to perform the experiments in this work.

### 3.5. Effect of Urea Concentrations

The influence of urea substrate concentration on AgNWs/HPMC/CS/Urease-modified electrode performance in heavy metal sensing applications was evaluated by cyclic voltammograms (CV) using 5 mM K_3_[Fe(CN)_6_]/0.1 KCl as a redox probe. The experiments were performed in the presence of 20 µM Hg(II), varying the final concentration of urea (2, 4, 6, and 8% *w*/*v*) and scanning from −600 to 600 mV with a scan rate of 100 mV (Figure 5). When the amount of urea as a substrate on the electrode surface increased, the peak current decreased gradually at 2% *w*/*v* of urea. With any further increase in the substrate amount, the current showed no further significant decrease. Thus 2% *w*/*v* urea was used for further experiments.

### 3.6. Effect of Reaction Time

The optimal response time, or incubation time, of the fabricated sensor with samples containing heavy metals is a key performance parameter. Fast sensing is preferable but sufficient time must be allowed for enzyme/heavy metal bonding and inhibition. The experiments were performed using the AgNWs/HPMC/CS/Urease-modified electrode in the presence of 20 µM Hg(II) as the inhibitor, 2% *w*/*v* urea, and 5 mM K_3_[Fe(CN)_6_/0.1 M KCl as a redox probe. These experiments were performed with the potential scanned from −600 to 600 mV with a scan rate of 100 mVs^–1^. The inhibition curve illustrated a slight increase in current for the first 10 min; the incubation time was monitored from 5 to 15 min (Figure 6). However, when the electrode was incubated with Hg(II) solution for 15 min, it did not show a further significant increase in the current. Thus, an incubation time of 10 min was used for further experiments.

### 3.7. Analytical Performance of the Urease Inhibition-Based Biosensor for Mercury (II) Detection

In this section, we investigate the AgNWs/HPMC/CS/Urease/SPCE biosensor quantitatively for the detection of Hg(II) ions. The modified electrode was investigated by incubation with various concentrations of Hg(II) ions in deionized water under optimal experimental conditions. The peak current due to ferricyanide was decreased as the concentration of Hg(II) increased, as shown in Figure 7a,b. This is attributed to an increase in the inhibition of free catalytic sites on the urease enzyme, which are available to the substrate. Under these conditions, the peak current decreased linearly with Hg(II) concentration over a linear range of the latter from 5 to 25 µM. A linear equation of I_pa_ = −(0.0313 ± 0.0054)x + 2.68 ± 0.07 could be fit to the data with a regression coefficient (R^2^) of 0.9267. The limit of detection (LOD) and limit of quantitation (LOQ) were found to be 3.94 μM (1069 ppb) and 6.50 μM (1765 ppb), respectively (the concentration of Hg(II) required to give the current change equal to 3 standard deviations (3σ; 10 replicate measurements of blank sample (I_0_) and 10 standard deviations (10σ) of I_0_ for LOQ).

### 3.8. Urease Inhibition-Based Biosensor for Mercury (II) Detection

In this work, AgNWs/HPMC/CS/Urease-modified electrode was used as a sensing electrode; the cyclic voltammograms (CVs) in the presence of 2% *w*/*v* urea for different concentrations of Hg(II) ions are shown in Figure 7a. In the absence of inhibitor, the redox peaks of ferricyanide were influenced by the presence of ammonium ions (NH_4_^+^) due to the reaction shown in Equation (1), an enzymatic reaction of urea with the urease immobilized on the sensing electrode [21,42]. This suggests that the AgNWs/HPMC/CS/Urease-modified electrode acquires positive charge as a result of the adsorption of ammonium ions in the enzymatic reaction process, and it attracts the negatively charged Fe(CN)_6_^4–^/^3–^ and further improves the electron transfer of the modified electrode.
(1)$$\mathrm{CO}{\left({\mathrm{NH}}_{2}\right)}_{2}+3H_{2}O\;\overset{\mathrm{Urease}}{\to }2{\mathrm{NH}}_{4}^{+}+{\;\mathrm{HCO}}_{3}^{-}+{\;\mathrm{OH}}^{-}$$
(2)$$\mathrm{Urease}+\mathrm{CO}{\left({\mathrm{NH}}_{2}\right)}_{2}\;\overset{\mathrm{Hg}\left(\mathrm{II}\right)}{\to }\;\mathrm{Urease}-\mathrm{Hg}\left(\mathrm{II}\right)$$

In the presence of a heavy metal ion such as Hg(II) as shown in Equation (2), an inactive enzyme–inhibitor complex is formed by the covalent bonding between the enzyme active center and the heavy metal ion via an irreversible reaction, as reported previously [21]. This causes a decrease in the production of ammonium ion (NH_4_^+^) species, leading to a decreased interaction between ferricyanide and the electrode and facilitating heavy metal sensing [21]. Moreover, the reactions in the presence of Hg(II) and varying pH conditions, ranging from 3 to 11 as shown in Appendix A, show that hydroxy ion species (OH^−^) may repel the negatively charged species (Fe(CN)_6_^4–/3–^), causing a reduction in electron transfer of the modified electrode and a significant decrease in anodic currents. Thus, in our experiments we selected pure DI water at a pH of about 6.4 to avoid this issue. We note that in the majority of real-world conditions, for example, mildly acidic natural waters or neutral human biofluids, the performance of our sensor was not degraded greatly if at all from the performance at pH 6.4.

### 3.9. Selectivity and Stability Studies

The selectivity of the modified electrode (AgNWs/HPMC/CS/Urease/SPCE) for the detection of Hg(II) compared its selectivity for other transition metal, alkaline metal, and alkaline earth metal ions (Cd(II), Cu(II), Fe(III), Ni(II), Pb(II), Zn(II), Co(II), Na(I), K(I), Ca(II), Mg(II)) was determined using cyclic voltammetry in the presence of 5 mM K_3_[Fe(CN)_6_/0.1 M KCl and 20 µM of each metal. The results shown in Figure 8a,b indicate that by far the greatest inhibition response was obtained for the presence of Hg(II) ions, confirming excellent selectivity. In addition, the stability test of modified SPCE was performed by storing electrodes in dry conditions at 4 °C and checking its sensing activity at seven-day intervals (Appendix A) in the presence of 5 mM K_3_[Fe(CN)_6_/0.1 M KCl and 20 µM Hg(II). The results indicated that even after 21 days, the current responses remained the same as the initial current response (first day).

### 3.10. Application of SPCE for the Determination of Hg(II) Ions in Real Water Samples

Next, the efficiency of the modified electrochemical Hg(II) biosensor was tested using real water samples where interferences with other ions may occur. Firstly, commercial drinking water was chosen for testing the performance of the biosensor and the results were validated with ICP-OES. The concentration of Hg(II) and the recovery percentage were calculated using the standard addition method by spiking known concentrations of Hg(II) ions (10 µM and 20 µM) in each sample. The results are depicted in Table 1, indicating the satisfactory recovery percentage of 101.62–105.26%. The results showed a good correlation with those obtained by ICP-OES, confirming the accuracy of the developed method for the detection of Hg(II). This suggested that this portable biosensor is a practical option for the determination of Hg(II) in real water samples.

### 3.11. Comparison with Other Procedures for Hg(II) Determination

As reported previously, Hg(II) is a strong carcinogen and very dangerous to human health and the environment. Excessive and prolonged exposure to this ion can cause various physical reactions (vomiting, headache, nausea, etc.) and lead to kidney malfunction and, eventually, death [4]. In 1972, the Joint FAO/WHO Expert Committee on Food Additives (JECFA) suggested that Hg(II) ions in the human body should be present at less than 3.3 μg/kg of body weight [43]. Moreover, the maximum acceptable limit of Hg(II) in drinking water specified by the WHO is 1 μg/L [17]. In the last decade, a variety of methods for the determination of Hg(II) at low ion concentrations have been reported, as depicted in Table 2. These methods include fluorescence spectroscopy, colorimetric techniques, and electrochemistry. In this research, our prepared disposable biosensor system has a higher limit of detection than that of other methods, but the linear range of our system is wider than that of other electrochemistry methods. While other methods with better LODs exist (Table 2), they can involve complicated instruments, lack portability, and require high sample volumes. Thus, our biosensor system is suitable for fast screening for high amounts of Hg(II) in the environment and real water samples. For example, it could be useful for fast detection of acute Hg(II) contamination in water spills in industrial contexts. In the future we will seek to improve the LOD of our sensor to reach the 1 ppb WHO limit, making the sensor much more useful in domestic drinking water contexts. Moreover, our sensor is portable and easy-to-use, thus making it suitable for point-of-care (POC) testing as well. In addition, we have started developing our sensor system to be used as a flat-card-sized electrochemical device using near-field communication (NFC) technology that connects it to the antenna of a smartphone, as described in Appendix A [41].

## 4. Conclusions

A facile fabrication of a AgNWs-modified SPCE electrode for a urease-based biosensor for rapid detection of Hg(II) ions was presented. AgNWs were selected for enhancing the sensor performance due to their inherent high conductivity and large active surface area. The sensor works well at room temperature, is cheap, and does not use any harmful materials. Thus, the sensor is disposable; however, it exhibits high stability over at least a month with a linear range of 5–25 µM for Hg(II) detection. The limit of detection (LOD) and limit of quantitation (LOQ) were observed under optimized conditions to be 3.94 µM and 6.50 µM Hg(II), respectively. Moreover, the biosensor was used to detect Hg(II) in real water samples, giving satisfactory results with a good percentage recovery, and the results were correlated with those obtained by ICP-OES. This sensor system may thus have potential for field-based rapid water safety determination.

## Data Availability

The research data supporting this publication are given within this paper and as Appendix A.

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
