# Peer review of "A Disposable Electrochemical Biosensor Based on Screen-Printed Carbon Electrodes Modified with Silver Nanowires/HPMC/Chitosan/Urease for the Detection of Mercury (II) in Water"

_biosensors, 2021, doi:10.3390/bios11100351_

Round 1

Reviewer 1 Report

Comments to the Authors:

Manuscript Number: 1362372

Title: A disposable electrochemical biosensor based on screen-printed carbon electrodes modified with silver nanowires/HPMC/chitosan/urease for the detection of mercury (II) in water

Major concerns:

The authors developed disposable and low-cost carbon screen-printed biosensor for detecting trace mercury (II) in water. Although being interesting and informative, I find that there are some major issues with the paper that require addressing prior to this being considered for publication in this journal. I have identified the main points for consideration below:

  • A major criticism of the work would be that very little was done to actually apply the method. Obtained results should be compared with those obtained with other reference/standard method(s).
  • Investigation about effect of pH toward biosensor can be very helpful to found optimal detection conditions of electrode. As you know, pH condition of water is varying by natural and pollution influences.
  • Please provide the calibration curves and explain how many electrodes were prepared and tested for each electrode to prepare the calibration curves?
  • I recommend that pictures 3a, b and c of the same magnification, eg 4 µm, be placed in the supplementary material, to make it easier to spot differences between differently modified surfaces.

Minor concerns:

There are some typos and omitted/added space in the text. The text of manuscript should be carefully checked.

Line 138 and 153

Equalize the writing of temperature, with or without space

Line 40 and 164

Equalize the writing of referemces, with or without space ([32], [33])

Line 218-220

Reformulate the sentence

“In this work, a nanocomposite material of AgNWs/HPMC/CS/Urease was coated 218 onto SPCEs by a simple casting method and cyclic voltammograms (CVs) measured (Fig. 219 2).”

Line 230

In word “After“ wrong font was used

Line 306

AgNWs/HPMC/ CS/Urease-

Table 1

Use uppercase letters for the first letter of the phrase

Reviewer 2 Report

The manuscript deals with the development of an electrochemical biosensor for Hg detection in drinking water by means of an inhibition process catalyzed by the metal itself. The overall level is good, and I really like the novelty of the proposed biosensor but there is a big issue with the LOD: the platform is able to detect Hg at a very big concentration, useless in drinking water samples, where the law limit by WHO is 1 ppb. I suggest changing the real sample to another one in which law limit fits with the data obtained with this platform.  A second possibility is to change the detection technique, with a more sensitive technique (SWV or DPV).

Besides this, some other suggestions and questions are reported:

  • Introduction: ref 22 is a review and does not exactly deal with “determination of the concentration of Hg(II) based on glucose oxidase and horseradish peroxidase (HRP), L-lactate dehydrogenase, and urease enzymatic inhibition have been constructed”. Please write the exact reference.
  • As law regulations are always written in ppb, to better understand the impact of the proposed study, please change the unit of the Hg metal, from M to ppb.
  • Please perform a study about the shelf life of the modified electrode, in order to understand their usability over the time.
  • Why did you use cyclic voltammetry instead of a more sensitive and qualitative technique such as SWV or DPV?
  • In chapter 3.2 please add an electrochemical impedance spectroscopy analysis to better characterize the surface.
  • Optimization performed in chapters 3.4, 3.5 and selectivity test in 3.7 should be carried out in triplicate, for statistical analysis of the data.
  • Chapter 3.5: for equation (1) please explain the inhibition deeply (reversible, irreversible...), and write down the equation better (see ref 21).
  • Ref 43 in the text does not deal with the sentence described "This causes a decrease of the ammonium ion (NH4+) species produced, leading to a decreased interaction between ferricyanide and the electrode and facilitating heavy metal sensing". Please correct it.
  • Chapter 3.4: please explain why there is this behavior at the electrode "When the amount of urea as a substrate on the electrode surface increases, the peak current decreases gradually". Why did you choose 2% if 8% gives you a higher current value? Figure 5b: Is this the anodic current value, right? Please write in the caption.
  • Please write the equation of the calibration curve using standard deviation.

Round 2

Reviewer 1 Report

The authors have significantly improved the manuscript. Thus I recommend this revised manuscript to be published in Biosensors.

Author Response

Thank you very much. 

Reviewer 2 Report

I feel satisfied with the answer, just can you please correct the significant figures in the equation of the calibration curve. 

Author Response

Reviewer Comment: I feel satisfied with the answer, just can you please correct the significant figures in the equation of the calibration curve.

Answer: Thank you very much for your comment. We already corrected the significant figures in the equation of the calibration curve and added to Fig. 7b to the lastest version of manuscript.